# Unveiling the Dynamics: Exploring User Affective and Behavioral Responses to Social Media

**DOI:** 10.3390/bs14070529

**Published:** 2024-06-25

**Authors:** Seonggoo Ji, Ihsan Ullah Jan

**Affiliations:** Department of Business Administration, Hanbat National University, 125, Dongseo-daero, Yuseong-gu, Daejeon 34158, Republic of Korea; sgji@hanbat.ac.kr

**Keywords:** social media demands, social media resources, social media fatigue, social media engagement, social media discontinuous intention, social media loyalty, social media psychology

## Abstract

Social media has outperformed traditional media as a source of interpersonal and masspersonal communication tools. The extant literature offers valuable knowledge on the positive and (or) negative attributes of social media and their ultimate effects on users’ affective and behavioral responses. However, it is unclear how the positive and negative attributes of social media affect users’ responses simultaneously. Drawing on the newly proposed social media demands and resources (SMD-R) model, the present study examined the positive and negative attributes of social media on the affective outcomes of social media fatigue (hereafter SM fatigue) and SM engagement (hereafter SM engagement) and behavioral outcomes of users in a single integrated framework. Data were gathered from 235 social media users in the Republic of Korea (hereafter Korea) to test the proposed framework. Partial least square structural equation modeling (PLS-SEM) was conducted, and the results showed that SM demands positively related to SM fatigue and negatively related to SM engagement. SM resources positively related to SM engagement and did not affect SM fatigue. SM fatigue positively related to SM discontinuous intention and negatively related to SM loyalty. Finally, SM engagement positively related to SM loyalty and negatively related to SM discontinuous intention. These findings contribute to the social media literature by proposing and empirically testing the SMD-R model, which integrates SM demands, SM resources, and the affective and behavioral responses of users.

## 1. Introduction

The unprecedented emergence and infusion of innovative technologies have influenced every sphere of human life, including human social interactions. One such example is the online social media platforms which generally comprise Meta (Facebook), Twitter, Instagram, Pinterest, and YouTube. These technologies are primarily computer-mediated communication tools for interpersonal and masspersonal communications. According to statistics, the number of social media users in 2020 was 3.91 billion, and it is forecasted the number of users will hit 6.05 billion by 2028 [1]. In another study, researchers reported that internet users across the world spend 147 min per day on social media platforms [2]. Such unprecedented emergence of social media platforms offers an avenue for researchers to study various aspects of social media and their ultimate effects on users’ attitudinal and behavioral responses. 

In this regard, the prior literature provides two main streams of studies about social media. First, researchers have examined the positive and negative attributes of social media on users’ positive attitudinal and behavioral outcomes, such as SM engagement [3] and social media word-of-mouth [4]. The second stream examines these attributes on user’s negative outcomes, such as sharing fake news [5], SM fatigue [2,6,7], SM addiction [8], technostress [9], and SM exhaustion [2,10,11,12]. However, a close evaluation of the social media literature shows that researchers have investigated the positive and negative factors of social media platforms, as well as their consequences, separately, and to the best of our knowledge, they have overlooked discussing the dual aspects of social media and their outcomes in a single integrated framework, ignoring the fact that users perceive both sides simultaneously. Recently, Sun et al. (2019) discussed the paucity of the literature and proposed the social demands–resources (SD-R) model by integrating the positive and negative attributes of social media in a framework [12]. However, this study had several limitations, as it overlooked examining the positive behavioral outcomes of the users along with the negative outcomes. Similarly, Sun et al. (2019) exclusively discussed demands based on social overload and resources based on social support, thus ignoring other positive and negative attributes of social media [12]. In past studies, researchers have argued that when consumers encounter innovative new products or services, they assess both the positive and negative reasons for adoption, which eventually influence users’ decision-making process [13]. Similarly, the customer demands–resources (CD-R) model postulates that when employees encounter customers during their interactions, positive customer aspects such as customer cognitive and emotional resources and negative aspects such as customer stressors influence the attitudinal and behavioral outcomes of the employees [14]. Thus, such a dichotomous model of both positive and negative factors provides a comprehensive understanding of human decision-making, attitude formation, and their ultimate behavioral consequences. Therefore, it is pertinent to examine how the positive and negative attributes of social media influence users’ affective and behavioral responses in a single integrated framework.

Drawing on the newly developed social media demands and resources (SMD-R) model, the current study attempts to explore the positive attributes as resources (e.g., helpfulness, playfulness, social values) and negative attributes as demands (e.g., information overload, communication overload, social overload) of social media and examine their affective (e.g., SM fatigue, SM engagement) and behavioral outcomes (e.g., SM discontinuous intention, SM loyalty) in a single framework. More specifically, this study seeks answer for the following research question: how do the demands and resources of social media influence users’ affective and behavioral responses? The SMD-R model is inspired by the job demands–resources (JD-R) theory and the customer demands–resources (CD-R) model [14,15,16]. These aforementioned theories are dichotomous in that they discuss not only the negative aspects (demands) but also the positive aspects (resources) of jobs and customers and, eventually, their effects on the employee’s attitudinal and behavioral outcomes in a single framework. The present study examines the positive influence of SM demands on users’ SM fatigue and the negative influence of SM demands on users’ SM engagement. Similarly, the present study investigates the positive effects of SM resources on SM engagement and the negative effects of SM resources on users’ SM fatigue.

The results of this study will advance the theory and offer insights for social media service providers. Theoretically, the current study proposes a social media demands and resources (SMD-R) model that encapsulates the interpersonal factors (e.g., social overload, helpfulness value) and masspersonal factors (e.g., information overload, communication overload) of social media in an integrated framework. By accomplishing this, this study contributes to the SM fatigue literature as well as the SM engagement literature by discussing the positive and negative attributes of social media. Managerially, this study offers insights to SM providers and practitioners to leverage factors of social media that play a vital role in determining user’s SM engagement and users’ SM fatigue. 

## 2. Theoretical Background and Hypotheses Development

### 2.1. SM Demands and Resources

To comprehensively examine the positive and negative attributes of social media, this study borrows a foundation from the job demands and resources (JD-R) theory. Primarily, this theory explains employees’ positive and negative psychological responses to job-related factors and their outcomes [15,16]. JD-R integrates four constructs, including demands, resources, strain, and engagement, to predict job outcomes [17]. Specifically, “job demands refer to those physical, psychological, social, or organizational aspects of the job that require sustained physical and/or psychological (cognitive and emotional) effort or skills and are therefore associated with certain physiological and/or psychological costs [strain]” (p. 312) [17]. Alternatively, job resources comprise “physical, psychological, social, or organizational aspects of the job that are either/or (a) functional in achieving work goals; (b) reduce job demands and the associated physiological and psychological costs; (c) stimulate personal growth, learning, and development” (p. 501) [16]. 

Drawing on the premises of JD-R theory, researchers developed the customer demands and resources (CD-R) model to understand customer-related positive and negative factors and their ultimate effects on customer satisfaction [14]. The CD-R model proposes that customer demands are the main drivers of frontline employees’ strains and customer resources are positive aspects that not only reduce employees’ strain but also positively influence customer-related outcomes [14]. Specifically, Stock and Bednarek (2014) defined customer demands as follows: “customer demands are the extent to which frontline employees encounter customers expressing negative behaviors whereas customer resources are the extent to which frontline employees perceive their customers as supportive of personal or work-related goals” (p. 402) [14]. Just like employees’ interactions with their jobs and customers, users of social media platforms also interact with social media and encounter positive and negative psychological experiences. Sun et al. (2019) proposed a social demands–resources model (SD-R) model based on the JD-R model [12]. Specifically, this study investigated social overload and social support as positive and negative aspects of social media and their effects on the user’s discontinuous intention. This study was primarily focused on discontinuous intention by exploring the effects of social overloads and social supports on social media exhaustion and engagement and their ultimate effects on social media discontinuous intention, and it overlooked examining positive outcomes in the framework. The current study proposes the SMD-R model, inspired by the aforementioned models (i.e., JD-R, CD-R, and SD-R), to understand the causal relationships of SM demands and resources on SM fatigue and SM engagement and their ultimate effects on discontinuous intention and loyalty. To be very specific, SM demands refer “to those social, psychological, and media aspects which demand the user’s psychological efforts and skills which in turn lead to strains for users”. On the contrary, SM resources refer to “the social, psychological, or media aspects which reduce the strains and are instrumental for personal growth and learning”. 

In past studies, researchers argued that the usage of usage media not only produces positive outcomes, such as social capital and psychological well-being, but it can also have negative outcomes, such as isolation, anxiety, and conflict [10,18]. Primarily, the negative outcomes happen because of overloads, which are the environmental stressors that exceed the individual’s optimal level of coping efforts. Generally, researchers in information technology have conceptualized overload into three elements, which are information, communication, and system feature overloads [18,19]. Generally, various studies have demonstrated that information overload, communication overload, and social overload are the major stressors for social media users [10]. Therefore, this study conceptualizes these overloads as SM demands. Similarly, in previous studies, researchers have given substantial attention to the negative elements or stressors of the usage of social media [10,11,20,21]. Therefore, as SM resources, the current study explores the positive elements of social media, which are the helpfulness value, playfulness value, and social value of social media usage. Researchers in previous studies have discussed that despite the stresses of social media, users want to use it because of its helpfulness [22], playfulness [23], and social value [24]. Hence, the current study explores these key positive aspects of social media platforms as SM resources and their effects on SM engagement and SM fatigue. The next section will discuss these elements of SM demands and SM resources in detail.

### 2.2. SM Demands

#### 2.2.1. Information Overload

As a component of SM demands, information overload is defined as an overwhelming amount of information that exceeds the user’s cognitive, physical, and/or psychological abilities to process and manage information effectively. In other words, users experience information overload if the received information exceeds their ability to process it [25,26,27,28]. Generally, during interpersonal and masspersonal communication, a tremendous volume of information is generated in the form of status updating, photo sharing, commenting, liking and sending emojis, and sending and receiving text and/or voice messages. 

#### 2.2.2. Communication Overload

Another element of SM demands is communication overload. Communication overload is defined as a situation wherein the need for communication in either an interpersonal situation or a masspersonal situation in social media exceeds the user’s cognitive, physical, and/or psychological abilities to continue the communication. Prior studies have shown that communication overload in social media distracts users’ attention and interferes with their behaviors [10,20,27,29]. Past studies have found that communication overload leads to negative outcomes such as emotional exhaustion and regret [10].

#### 2.2.3. Social Overload

The phenomenon of social media has become ubiquitous, and users can interact with people anywhere and anytime to fulfill the needs of interpersonal or masspersonal social interactions. Specifically, social media social overload is defined as a situation wherein the need for social interaction on social media exceeds the user’s cognitive, physical, and/or psychological abilities to continue the social relationships. For instance, users perceive “that there are too many social requests to be processed and superabundant social support requests to be fulfilled on social network sites” [10,20]. In the prior literature, a study argued that social overload leads to negative outcomes such as exhaustion [30].

### 2.3. SM Resources

#### 2.3.1. Helpfulness Value

Social media helpfulness value refers to “the extent to which users gain resources and helpful information from their exploration of social media sites” (p. 152) [22]. In the prior literature, different theoretical frameworks were drawn in examining the determinants of social media usage for social interactions, particularly in interpersonal interactions. In this regard, the key theories are social capital [31,32] and gratification theory [33]. In line with these aforementioned studies, Bright et al. (2015) documented the significance of helpfulness as an important component of social media usage [22]. 

#### 2.3.2. Playfulness Value

Social media playfulness value refers to the arousal of users’ positive feelings that is triggered by social media. Correspondingly, playfulness is a hedonic component of social media consumption. In the prior literature, Kaur et al. (2018) found that playfulness as an emotional value led to social media usage intention [23]. Similarly, for social media communities on masspersonal levels, hedonic needs and enjoyment influence consumers to use social media [34,35]. Thus, playfulness is a very important element of social media which is instrumental as a positive factor for the usage of social media. 

#### 2.3.3. Social Value

Social value refers to the utility that is derived from associations with family, friends, and the community at large by using social media for interpersonal and masspersonal communications. This social value expectation is a major reason to indulge in social networking [31,36]. Social media features like social interaction and social enhancement help users to develop connections with new acquaintances who share similar interests [23]. Similarly, Raacke and Bonds-Raacke (2008) argued that maintaining an interpersonal relationship is the ultimate goal of users of social media [33]. Yang and Chien-Liang (2014) stated that in terms of social values, social media provide a platform for users to build interpersonal networks by offering various communication tools, such as text messages, visuals, emoticons, and voice notes [24]. 

### 2.4. SM Demands, SM Fatigue, and Social Media Engagement

SM fatigue is defined as a “subjective and self-evaluated feeling of tiredness from social media usage” [26]. More precisely, Ravindran et al. (2014) defined SM fatigue as “a subjective, multi-dimensional user experience comprised of feelings such as tiredness, annoyance, anger, disappointment, guardedness, loss of interest, or diminished need/motivation associated with various aspects of social media use and interactions” (p. 2317) [37]. In the prior literature, researchers have examined the antecedents of SM fatigue, such as social media overloads [26]. Lee et al. (2016) demonstrated that information overload, communication overload, and system feature overload increase SM fatigue [26]. Similarly, Bright et al. (2015) held the view that SM fatigue happens because of interpersonal interactions with friends and acquaintances, which usually lead to online social comparison and self-disclosures [22]. Likewise, technological aspects of social media also predict SM fatigue because of the added features. In past studies, researchers have discussed SM fatigue by either investigating the effects of positive factors or negative determinants [26,38]. This study explores the effects of both positive and negative influences of SM demands and SM resources on SM fatigue.

Past studies have argued that demands, such as job demands and customer demands, produce negative psychological strains on employees and customers [14,16,39]. Few studies have documented how these job demands affect the job fatigue of employees. Generally, users of social media are overwhelmed and lose their cognitive control because of information, social, and communication overloads [2,10,12,20,25,30]. Thus, SM demands might produce a psychological strain on users. Specifically, researchers have found that social media overload, which often comprises information, communication, and social overloads, led to users’ social media exhaustion, which consequently positively affected discontinuous usage [11,12,38]. Based on these findings, unlike the previous studies, the current study argues that SM demands, which are social media-related stressors, demand the user’s physical and/or psychological efforts and skills, which are instrumental in SM fatigue. In contrast, these SM demands will have a negative effect on SM engagement because of their overwhelming information, social, and communication-related stressors. Hence, we propose the following:

**H1.** 
*SM demands positively affect user’s SM fatigue.*


**H2.** 
*SM demands negatively affect users’ SM engagement.*


### 2.5. SM Resources, SM Demands, and SM Engagement

Engagement has been extensively studied in the prior literature [40,41]. In earlier studies, the researchers have focused more on the traditional engagement process between companies and customers. Recently, due to the penetration of digital technologies into customer–firm relationships, researchers have started examining the impacts of these new technologies on marketing strategies [40,42]. Specifically, in the context of social media, engagement is conceptualized into three dimensions of engagement, including cognitive, affective, and behavioral components. According to Cao et al. (2021), cognitive engagement refers to “being akin to the mental processes involved in focusing on intense attention and absorption” (p. 836) [3]. Similarly, affective engagement refers to “emotional reactions, such as enthusiasm and enjoyment”; whereas behavioral engagement is “the active manifestations of the engagement concept which include sharing, learning and endorsing behaviors” (p. 836) [3]. In the prior literature, researchers have mainly discussed SM engagement as SM engagement behavior [3,43,44], customers’ SM engagement with brand communities [38,40], and students’ SM engagement [45]. Therefore, the current study will advance the examining the demands and resources on SM engagement of the users.

As discussed above, SM resources are the positive aspects of social media, which reduce psychological strains and increase social media usage. Although the negative aspects discourage users from using it, the positive aspects, such as social media helpfulness, social media playfulness, and social media social values, are instrumental in its usage [22,23]. These positive aspects of social media are conceptualized as SM resources. Generally, resources such as job resources and customer resources reduce negative psychological strains and increase positive outcomes [14,17]. Hence, the current study infers and proposes that SM resources affect SM engagement positively and SM fatigue negatively. Thus, the following are hypothesized:

**H3.** 
*SM resources positively affect users’ SM engagement.*


**H4.** 
*SM resources negatively affect users’ SM fatigue.*


### 2.6. SM Fatigue, SM Discontinuous Intention, and SM Engagement

Social media discontinuous intention refers to an “individual’s intention to change his or her system use status quo by either reducing the intensity of his or her social media use, temporarily or permanently discontinuing use of social media altogether, or switching to another social media platform” [46]. Specifically, researchers have argued that users showed their discontinuous usage intention toward social media not only by abruptly ceasing their usage but by reducing usage intensity [47], permanent or temporary discontinuance [48,49], and switching to another network [50]. In past studies, researchers have extensively given attention to exploring the effects of negative factors on discontinuous intention, such as exhaustion [10], regret [10], privacy concerns [51], dissatisfaction [29], and maladaptive coping strategies [20]. Unlike the previous studies, the current study explores discontinuous intention by examining SM fatigue and SM engagement as antecedents to SM discontinuous intention based on the newly proposed SM demands and resources model. 

In the prior literature, researchers explored the role of SM fatigue on SM discontinuous intention [29,40,46]. Surprisingly, a large number of these studies have been conducted and there are mixed findings. Therefore, the present study explores the positive relationship between SM fatigue and SM discontinuous intention. In line with the aforementioned findings, it is inferred that SM fatigue leads to SM discontinuous intention. This is because users suffer from mental and psychological exhaustion when they expose themselves to social media for a longer period of time [6]. Consequently, the tiredness and mental and psychological exhaustion of users lead to discontinuous intention. In contrast, it is likely that the tiredness and mental and psychological exhaustion of users reduce the users’ SM engagement. Based on these arguments, we propose the following:

**H5.** 
*SM fatigue positively affects users’ SM discontinuous intention.*


**H6.** 
*SM fatigue negatively affects users’ SM engagement.*


### 2.7. SM Engagement, SM Loyalty, and SM Discontinuous Intention

Surprisingly, in the prior literature, loyalty has been discussed in the context of the social media community, and researchers have overlooked discussing users’ SM loyalty directly. In the current study, SM loyalty refers to users’ loyal behaviors, such as users’ positive attitude toward social media platforms, users’ tendency to say positive things about social media, and encouraging family and friends to use social media. Just like customer loyalty, which often comprises both attitudinal and behavioral aspects [52], SM loyalty can have both of these components. Attitudinal loyalty refers to a favorable attitude of users towards social media, whereas behavioral loyalty refers to a continuous re-usage of social media. In past studies, the behavioral and attitudinal dimensions of customer loyalty have been expressed as “composite measurements”, providing a foundation for understanding loyalty [46]. Hence, the current study explores SM loyalty based on composite measurements. 

Social media engagement is a key determinant of behavioral intention, word-of-mouth [53], brand trust [40], and social media identification [54]. The positive relationship between engagement and loyalty is well established in the marketing literature [38,55], yet researchers have overlooked examining this relationship in the context of social media. Nevertheless, a few researchers have found that SM engagement leads to loyalty to the community on the social networking platform [56,57]. Lim et al. (2015) argued and found that the higher the SM engagement of the users, the higher the sports-channel loyalty [56]. Similarly, the findings of Molinillo et al. (2020) established that customer engagement with a website increases the loyalty of the users of the website [57]. Moreover, Sun et al. (2019) argued that high social media engagement of users decreases their discontinuous intentions toward social media [12]. Since the aforementioned studies validate the role of engagement in loyalty, it is argued that the SM engagement of users increases their loyalty and decreases their discontinuous intention. Thus, the following are proposed:

**H7.** 
*SM engagement positively affects users’ SM loyalty.*


**H8.** 
*SM engagement negatively affects users’ SM discontinuous intention.*


Drawing on the JD-R theory, the new conceptual framework of SMD-R in Figure 1 summarizes the relationship between SM demands, SM resources and the user’s affective and behavioral outcomes.

## 3. Method

### 3.1. Sample and Data Collection

An online questionnaire-based survey was conducted in Korea to gather data from social media users. Korea offers an ideal context to test the proposed relationships. According to the latest data, the number of social media users in Korea will reach 47.75 million by 2026, making up 90% of the total population [58]. Thus, the number of social media users in Korea is significantly high, as people use social media for both interpersonal and masspersonal communications. Therefore, examining the positive and negative aspects of social media and their ultimate consequences on users’ attitudinal and behavioral responses is pertinent.

Researchers contacted the marketing research firm H-Research to collect users’ responses regarding social media. H-Research is one of the biggest research firms, maintaining a panel of approximately 2.9 million respondents, representing the entire population from across the country. A total of 250 responses were obtained from social media users regarding the positive and negative attributes of social media and their respective affective and behavioral responses. After the preliminary analysis, 15 responses were eliminated because of missing data and outlier issues, and a total of 235 were finalized to conduct the formal analysis. 

The frequency distribution of the sample shows that in terms of gender, 124 were males, which makes 52.8 percent of the total sample, whereas 111 were females, which makes 47.2 percent of the total sample. In terms of age group, respondents aged in their 50s and above were the highest with 75 participants, making 31.9 percent of the total sample, followed by the age group in their 40s with 66, making 28.1 percent of the total sample. All respondents in the final sample reported that they knew social media and used social media on a daily basis. A total of 30 percent of users reported 1–3 h of daily social media usage, with 37 percent spending 1 h every day. Table 1 provides the details of the sample.

### 3.2. Measurements

To measure the key constructs of the study, scales were adapted from the prior literature, and a few items were modified to fit in the current context. Information overload with 3 items, communication overload with 5 items, and social overload with 5 items were specifically adapted from the study of Cao and Sun (2018) [10]. The helpfulness value of social media was measured with 3 items from the study of Bright et al. (2015) [22]. Playfulness was assessed with 3 items and social value was assessed with 3 items from the study of Kaur et al. (2018) [23]. SM fatigue was measured with 5 items based on the scale of Islam et al. (2021) [19]. SM engagement was 4 items assessed on the scale of Lim et al. (2015) [56]. SM loyalty was measured with 4 items based on the study of Nisar et al. (2016) [46], and finally, SM discontinuous intention was assessed with 3 items based on Cao and Sun (2018) [10]. All the responses from the users were assessed on a 5-point Likert scale of “1 (strongly disagree) to 5 (strongly agree)”. All the measurement items are given in detail in Table 2.
behavsci-14-00529-t002_Table 2Table 2Measurement scales, reliability, and validity assessment.ConstructsLabelsLoadingsRho-ACRAVESourceInformation overload (α = 0.97)“I am often distracted by the excessive amount of information in SM”.0.970.970.980.94Cao and Sun (2018) [10]“I am overwhelmed by the amount of information that I process on a daily basis from SM”.0.97


“I feel some problems with too much information in SM to synthesize instead of not having enough information”.0.96


Communication overload(α = 0.96)“I receive too many messages from friends through SM”.0.930.960.970.87Cao and Sun (2018) [10]“I feel as if I have to send more messages to friends through SM than I want to send”.0.93



“I feel that I generally receive too many notifications on new postings, push messages, and news feeds, among others from SM as I perform other tasks”.0.93



“I often feel overloaded with SM communication”.0.93




“I receive more communication messages and news from friends on SM than I can process”.0.93



Social overload (α = 0.97)“I take too much care of the well-being of my friends on SM”.0.930.970.980.91Cao and Sun (2018) [10]“I deal with my friends’ problems on SM too much”.0.95


“My sense of responsibility for how much fun my friends have on SM is too strong”.0.96


“I care for my friends on SM too often”.0.96


“I pay too much attention to my friends’ posts on SM”.0.96


Helpfulness value (α = 0.87)“SM helps me to keep in touch with family and friends”.0.900.880.920.80Bright et al. (2015) [22]“SM helps me to learn new things”.0.90



“SM helps me to do my tasks”.0.87



Playfulness value (α = 0.96)“SM helps me to have fun”.0.930.960.970.90Kaur et al. (2018) [23]“SM offers excitement to me”.0.89


“SM offers an enjoyable experience to me”.0.94


Social value(α = 0.91)“Using SM enhances my reputation among friends”.0.920.910.940.84Kaur et al. (2018) [23]“Using SM can help me impress others”.0.93


“Using SM can help me feel important”.0.90


SM fatigue (α = 0.97)“I find it difficult to relax after continually using SM”.0.930.970.970.88Islam et al. (2021) [19]“After a session of using SM, I feel really fatigued”.0.94“Due to using SM, I feel rather exhausted”.0.95“After using SM, it takes effort to concentrate in my spare time”.0.94“During SM use, I often feel too fatigued to perform other tasks well”.0.95SM engagement (α = 0.96) “I post my feelings in real-time on SM”.0.950.960.970.90Lim et al. (2015) [56]“I post my feelings when I like/dislike something on SM”.0.93


“I quote from the SM when something is good or witty”.0.95


“I express my feelings about anything on SM”.0.96


SM loyalty (α = 0.95) “I would say positive things about SM”.0.920.950.960.86Nisar et al. (2016) [46]“I intend to keep using SM”.0.91


“I intend to recommend SM to my friends”.0.95


“I would encourage relatives and friends to use SM”.0.94


SMdiscontinuousintention (α = 0.95)“I intend to stop using SM in the next three months”.0.960.950.970.91Cao and Sun (2018) [10]“I will drastically cut down on SM use in the next three months”.0.95


“I will not be using SM in the next three months”.0.95





Exploratory factor analysis indicated a three-factor solution—information, communication, and social overload for SM demands. The Kaiser–Meyer–Olkin value was 0.95, and Barlett’s test of sphericity was significant (*p* < 0.01), which confirmed the sample’s adequacy for factor analysis. The factor loadings for information overload were reported between 0.82 and 0.84; for communication overload, they were 0.76–0.84; and for social overload, 0.80–0.84. Similarly, exploratory factor analysis was conducted for SM resources. The Kaiser–Meyer–Olkin value was 0.92, and Barlett’s test of sphericity was significant (*p* < 0.01), confirming the sample’s adequacy for factor analysis. Two of the items of helpfulness were cross-loaded with playfulness, and one item had lower loading and was therefore eliminated, and the remaining factor loadings were between 0.79 and 0.84, while playfulness had factor loadings of 0.76–0.87 and social value had factor loadings of 0.82–0.91.

We examined the psychometric characteristics of the scale by conducting confirmatory factor analyses (CFAs) for both of the newly developed scales for SM demands and resources. Following the generally accepted recommendation for unidimensionality tests, a test of chi-square differences was conducted to compare the higher-order construct with the unidimensional construct [59,60]. A three-factor model with information overload, communication overload, and social overload as discrete latent variables was conducted for SM demands. Several fit indices for the three-factor model were better (X^2^ = 2.05, GFI = 0.92, CFI = 0.99, TLI = 0.98, IFI = 0.99, NFI = 0.97, RMSEA = 0.07, RMR = 0.05) than the one-factor model (X^2^ = 19.52, GFI = 0.45, CFI = 0.73, TLI = 0.67, IFI = 0.73, NFI = 0.72, RMSEA = 0.28, RMR = 0.15). Similarly, we repeated the procedure for SM resources. A three-factor model with helpfulness, playfulness, and social value as discrete latent variables for SM resources was conducted. The results of the three-factor model were better (X^2^ = 2.68, GFI = 0.94, CFI = 0.97, TLI = 0.96, IFI = 0.97, NFI = 0.96, RMSEA = 0.09, RMR = 0.05) than the one-factor model (X^2^ = 17.89, GFI = 0.68, CFI = 0.71, TLI = 0.65, IFI = 0.71, NFI = 0.70, RMSEA = 0.27, RMR = 0.12). Similarly, the convergent validity of the scale was confirmed, as the standard factor loadings were above the cutoffs of 0.70 for information overload (0.94–0.96), communication overload (0.88–0.93), social overload (0.90–0.96), helpfulness value (0.81–0.84), playfulness value (0.82–0.92), and social value (0.81–0.92). The values of average variance extracted (AVE) and the composite reliability (CR) were above the benchmarks of 0.50 and 0.70, respectively.

### 3.3. Data Analysis and Results

To examine the measurements and structural model, PLS-SEM was applied with SmartPLS software 3.0. Researchers studying management are becoming more interested in PLS-SEM, a method that has been around for a while and is applied to examine structural models [61]. Specifically, PLS-SEM is a composite-based analysis approach, which is often preferred in the context of complex research model estimation [62,63,64]. Based on the recommendation of Kock (2015), the common method bias (CMB) was examined based on VIF scores [65]. We found that all VIF values for the latent variables were below the cutoff value of 3.3 [66], confirming the absence of CMB. PLS-SEM analysis was conducted systematically. First, the researchers evaluated the psychometric characteristics of the measurement model, followed by structural model assessment. 

## 4. Results 

### 4.1. Measurement Model Assessment

All the loadings of measurement items were significant (with coefficients above 0.70) on their respective constructs, and the AVE was above 0.50 [62,67]. Table 2 reports that the factor loadings were above 0.70 (*p* < 0.01), and the AVE was above the cutoff of 0.50 [68], which confirms the convergent validity. To validate the reliability, the values of composite reliability, rho_A, and the Cronbach alpha coefficient were evaluated [62]. The findings indicated that the coefficients of Cronbach Alpha (α), CR, and rho_A were above the recommended value of 0.70 [62,68], thus confirming the reliability of the constructs.

Similarly, Fornell and Larcker’s (1981) approach was used to evaluate the discriminant validity [67]. The findings showed that the squared roots of the AVE exceeded the inter-correlation coefficients. Table 3 shows the correlation matrix and the square roots of the AVE in bold.

In this study, SM demands and SM resources are second-order constructs. Therefore, we initially analyzed the psychometric properties of the constructs before conducting a structural model estimation. The results of the confirmatory factor analysis for the second-order constructs are given in Table 4.

The results of the higher-order constructs showed that all the psychometric properties of SM demands and SM resources fulfilled the minimum thresholds in that the values of all latent constructs had AVE coefficients of α, CR, and CR that were more than the relevant thresholds of 0.70, 0.70, and 0.50, respectively [68].

### 4.2. Structural Model Assessment

The structural model of the study were assessed in light of the directions of [62]. Although there are relatively fewer indices for model fitness testing in PLS-SEM than in CB-SEM, the Standardized Root Mean Square Residual (SRMR) and the Normal Fit Index (NFI) are often reported to assess model fitness. The results showed that the SRMR of the structural model was 0.05 less than the cutoff of 0.08, and the NFI was 0.89, which was acceptable. Table 5 reports the indices for model fitness.

Table 6 provides the results of the hypotheses testing in detail. The results showed that SM demands had increased SM fatigue (β = 0.88, *p* < 0.01), whereas SM demands negatively affected SM engagement (β = −0.15, *p* < 0.01); thus, H1 and H2 of the study were supported. SM resources had a significant positive effect on SM engagement (β = 0.67, *p* < 0.01), whereas the negative effect of SM resources on SM fatigue was not supported (β = 0.01, *p* > 0.05); thus, H3 of the study was supported. Hypothesis 5 suggested that respondents with higher social fatigue would have a high SM discontinuous intention. The data indicated a positive and significant relationship between SM fatigue and SM discontinuous intention (β = 0.73, *p* < 0.01); thus, Hypothesis 5 of the study was supported. The results revealed that SM fatigue had a significant negative effect on SM loyalty (β = −0.15, *p* < 0.01), which supported the proposed Hypothesis 6 of the study. Finally, SM engagement related to SM loyalty positively (β = 0.85, *p* < 0.01) and related to SM discontinuous intention negatively (β = −0.16, *p* < 0.01); thus, H7 and H8 of the study were supported.

The results of the coefficient of determination (R^2^) and predictive relevance (Q^2^) are reported in Table 7. The results of the R^2^ showed that SM demands explained 77% of the variance in SM fatigue, and SM demands and resources explained 61% of the variance in SM engagement. Similarly, SM fatigue and SM engagement explained 68% of the variance in SM discontinuous intention and 88% in SM loyalty. Thus, except for SM engagement, which was explained moderately (R^2^ = 0.33), all the endogenous constructs showed a substantial effect size of R^2^ > 0.67 [69]. To assess the predictive relevance, we conducted a blindfolding analysis in SmartPLS with an omission distance of 7. The Q^2^ results indicated that the model had strong predictive relevance, as the values of SM fatigue, SM engagement, SM discontinuous intention, and SM loyalty were above the cutoff of Q^2^ > 0.35 [70].

### 4.3. Ad hoc Mediation Analysis

We conducted an ad hoc mediation analysis to further explain the nature of the relationship between SM antecedents and the consequences of SM fatigue and SM engagement. The results showed that SM demands had an indirect effect on SM discontinuous intention through SM fatigue (β = 0.57, *p* < 0.01), whereas the direct effect was insignificant (β = 0.11, *p* > 0.05); thus, SM fatigue fully mediated the relationship. The indirect effect of SM demands on SM loyalty through SM fatigue was significant (β = −0.11, *p* < 0.01), whereas the direct effect was not significant (β = −0.01, *p* > 0.05); thus, SM fatigue fully mediated the relationship. 

The indirect effect of SM demands on discontinuous intention through SM engagement was significant (β = 0.02, *p* < 0.05), whereas the direct effect was not significant (β = 0.11, *p* > 0.05); thus, SM engagement fully mediated the relationship. The indirect effect of SM demands on SM loyalty through engagement was significant (β = −0.10, *p* < 0.05), whereas the direct effect was not significant (β = −0.01, *p* > 0.05); thus, SM engagement fully mediated the relationship. 

Finally, the indirect effect of SM resources on SM discontinuous intention through SM engagement was significant (β = −0.10, *p* < 0.01), whereas the direct effect was not significant (β = −0.01, *p* > 0.05); thus, SM engagement fully mediated the relationship. The indirect effect of SM resources on SM loyalty was significant through SM engagement (β = 0.52, *p* < 0.01), and the direct effect was significant too (β = 0.16, *p* < 0.01); thus, SM engagement partially mediated the relationship. Table 8 summarizes the results. 

## 5. Discussion and Implications 

### 5.1. Theoretical Implications

Theoretically, this study has proposed a SM demands and resources model (SMD-R) which discusses the determinants and outcomes of SM affective and behavioral responses in a single framework. Previous studies have either examined positive attitudinal and behavioral outcomes, such as SM engagement [3] and social media word-of-mouth [4], or negative attitudinal and behavioral outcomes, such as SM fatigue [2,6,7], SM addiction [8], technostress [9], and SM exhaustion [2,10,11,12]. However, this study has proposed an SMD-R model that encapsulates the interpersonal factors (e.g., social overload, helpfulness value) and masspersonal factors (e.g., information overload, communication overload) of social media in an integrated framework. Specifically, SM demands are a high-order construct comprising information overload, communication overload, and social overload. Unlike previous studies, the current study has explored the effects of SM demands not only on SM fatigue but on SM engagement too. Hence, the findings advance the literature on SM engagement by exploring the negative determinant of SM engagement. In other words, SM demands decrease users’ SM engagement.

Similarly, unlike previous studies which focused more on social media’s negative aspects and their effects on psychological strains such as SM fatigue [2,6,7], the present study examined SM resources such as helpfulness value, playfulness value, and social value and found that SM resources decrease SM fatigue. Likewise, drawing on the SMD-R model, the present study has contributed to the SM engagement literature by empirically testing the positive influence of SM resources. Finally, based on SMD-R, the current study has advanced the literature on SM discontinuous intention and SM loyalty by exploring the roles of SM fatigue and SM engagement. 

Finally, based on SMD-R, the current study has advanced the literature on SM discontinuous intention and SM loyalty by exploring the roles of SM fatigue and SM engagement. The current study showed that SM fatigue has positive effects on SM discontinuous intention, but it has negative effects on SM loyalty. In contrast, SM engagement is related positively to SM loyalty and negatively to SM discontinuous intention.

### 5.2. Managerial Implications

Managerially, the findings of this study offer insights to social media providers and users by discussing the positive and negative factors of SM fatigue and SM engagement, respectively. First, the results showed that SM demands affected SM fatigue positively and SM engagement negatively. Therefore, social media providers should devise strategies to alleviate SM demands. In this regard, the manager can “allow users to set limits on communication or social requests and autonomously choose to display or hide posts or provide content management functions, such as displaying the abstract of an article and classifying content based on users’ interests” [20]. Similarly, users can also avoid SM fatigue by using the option for “disabled notifications” that restricts all the incoming notifications of new incoming information, new post and picture notifications, and notifications for text or voice calls. 

Second, the results showed that SM resources mitigate SM demands; therefore, the providers of social media companies and users can enhance the positive aspects of social media, such as social media helpfulness, social media playfulness, and social media social values. In this regard, social media companies should encourage a sense of community through social media, which can enhance users’ exchange of helpful and enjoyable moments together. These kinds of resources not only enhance the SM engagement of users but can also mitigate SM fatigue and social media stressors. 

Third, based on the SMD-R model, the results showed that SM fatigue has a positive effect on SM discontinuous intention and a negative effect on SM loyalty. Therefore, the social media provider should consider mitigating social fatigue by restricting the aforementioned overloads, including information, communication, and social overloads. Moreover, social media providers can decrease discontinuous intentions by increasing SM engagement. Finally, social media providers can increase SM loyalty by focusing more on the SM engagement of users.

### 5.3. Limitations and Future Research Directions

The following are a few limitations which should be addressed in future studies. First, this study is the first attempt to discuss SM demands and SM resources and their effects on SM fatigue and SM engagement in the single integrated framework of the SMD-R model. By doing so, the current study has examined the dichotomous factors of key positive and negative aspects and their ultimate outcomes on users’ SM fatigue, SM engagement, and SM discontinuous intention, as well as SM loyalty. However, the study has overlooked examining these relationships from the perspectives of different users, such as by considering personality differences, differences in social media usage intensity, and social media addiction. Therefore, future studies can explore the framework by discussing the moderating effects of personality traits, social media addiction, and the sense of users’ fear of missing out.

Second, the researchers have used a self-rated cross-sectional study to test the proposed framework of the SMD-R model; therefore, this study has limitations regarding the generalizability of the model. Hence, researchers can test this newly proposed model of SMD-R by using alternate research methodologies such as longitudinal studies. 

Finally, in order to empirically validate the newly proposed framework of the SMD-R model, the researchers gathered data in Korea; therefore, to generalize the results, researchers can replicate this framework in other countries.

## Figures and Tables

**Figure 1 behavsci-14-00529-f001:**
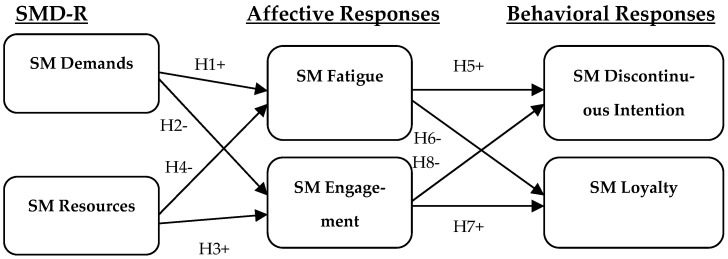
Research model.

**Table 1 behavsci-14-00529-t001:** Sample profile.

Demographic Characteristics	Frequency	Percentage
Average time spent on social media		
1 h daily	79	33.6%
1–3 h daily	90	38.3%
>3 h daily	38	16.2%
1 h weekly	12	5.1%
3–4 h weekly	11	4.7%
>3 h weekly	5	2.1%
Gender		
Male	124	52.8%
Female	111	47.2%
Age		
20–29	51	21.7%
30–39	43	18.3%
40–49	66	28.1%
50–59	75	31.9%
Marital status		
Married	144	61.3%
Un-married	85	36.2%
Other	6	2.5%
Education		
High school diploma	23	9.8%
Enrolled in bachelor degree	166	70.6%
Bachelor degree	2	0.9%
Enrolled in graduate school	27	11.5%
Postgraduate degree	23	9.8%
Total	235	100%

**Table 3 behavsci-14-00529-t003:** Correlation matrix and square roots of AVE.

	1	2	3	4	5	6	7	8	9	10
1. CO	**0.93**									
2. PV	−0.46	**0.92**								
3. IO	0.73	−0.55	**0.97**							
4. SO	0.77	−0.52	0.74	**0.95**						
5. SDI	0.65	−0.5	0.69	0.71	**0.95**					
6. SME	−0.41	0.73	−0.56	−0.46	−0.52	**0.95**				
7. HV	−0.43	0.68	−0.52	−0.47	−0.42	0.61	**0.89**			
8. SML	−0.47	0.76	−0.63	−0.52	−0.59	0.93	0.64	**0.93**		
9. SMF	0.79	−0.48	0.78	0.83	0.81	−0.49	−0.43	−0.58	**0.94**	
10. SV	−0.22	0.53	−0.38	−0.27	−0.29	0.56	0.52	0.57	−0.30	**0.92**

Notes: Bold scores show the squared roots of AVE, CO: communication overload, PV: playfulness value, IO: information overload, SO: social overload, SDI: social media discontinuous intention, SME: social media engagement, HV: helpfulness value, SML: social media loyalty, SMF: social media fatigue, SV: social value.

**Table 4 behavsci-14-00529-t004:** Higher-order constructs.

Constructs	Items	Std. Factor Loadings	Cronbach’s Alpha	Rho_A	CR	AVE
SM demands	IO	0.91	0.90	0.90	0.94	0.83
	CO	0.91				
	SO	0.92				
SM resources	HV	0.90	0.80	0.83	0.88	0.72
	PV	0.87				
	SV	0.77				

IO: information overload, CO: communication overload, SO: social overload, HV: helpfulness value, PV: playfulness value, SV: social value.

**Table 5 behavsci-14-00529-t005:** Model fit summary.

Fitness Indices	Saturated Model	Estimated Model
SRMR	0.05	0.05
d-ULS	0.52	0.61
d-G	0.54	0.55
Chi-Square	727.50	718.96
NFI	0.89	0.89

**Table 6 behavsci-14-00529-t006:** Results of PLS-SEM.

Relationship		Estimate	t-Value	*p*-Value	Results
H1	SM demands → SM fatigue	0.88	35.78	0.00	Supported
H2	SM demands → SM engagement	−0.15	2.82	0.00	Supported
H3	SM resources → SM engagement	0.67	12.84	0.00	Supported
H4	SM resources → SM fatigue	0.01	0.20	0.84	Not supported
H5	SM fatigue → SM discontinuous intention	0.73	20.65	0.00	Supported
H6	SM fatigue → SM loyalty	−0.15	5.62	0.00	Supported
H7	SM engagement → SM loyalty	0.85	37.66	0.00	Supported
H8	SM engagement → SM discontinuous intention	−0.16	3.93	0.00	Supported

**Table 7 behavsci-14-00529-t007:** Results of R^2^ and Q^2^.

Endogenous Latent Constructs	R^2^	Q^2^
SM fatigue	0.77	0.67
SM engagement	0.61	0.54
SM discontinuous intention	0.68	0.62
SM loyalty	0.88	0.76

**Table 8 behavsci-14-00529-t008:** Mediation analysis results.

Relationships	Direct Effect			Indirect Effect				
	β	t	*p*	β	t	*p*	5.0%	95.0%
SM demands → SMF → SDI	0.11	1.23	0.10	0.57	7.93	0.00	0.45	0.69
SM demands → SMF → SML	−0.01	0.19	0.43	−0.11	4.90	0.00	−0.15	−0.08
SM demands → SME → SDI	0.11	1.23	0.10	0.02	1.83	0.03	0.01	0.04
SM demands → SME → SML	−0.01	0.19	0.43	−0.10	2.43	0.01	−0.16	−0.03
SM resources → SMF → SDI	−0.01	0.15	0.44	0.01	0.05	0.48	−0.04	0.04
SM resources → SMF → SML	0.16	4.25	0.00	−0.00	0.05	0.48	−0.01	0.01
SM resources → SME → SDI	−0.01	0.15	0.44	−0.10	3.64	0.00	−0.15	−0.06
SM resources → SME → SML	0.16	4.25	0.00	0.52	11.97	0.00	0.44	0.58

SMF: SM fatigue, SME: SM engagement, SML: SM loyalty, SDI: SM discontinuous intention.

## Data Availability

Data will be available on request.

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
