# Peer review of "Unveiling the Dynamics: Exploring User Affective and Behavioral Responses to Social Media"

_behavsci, 2024, doi:10.3390/bs14070529_

Round 1

Reviewer 1 Report

Comments and Suggestions for Authors

Thank you for the invitation to review this interesting paper entitled „Unveiling the Dynamics: Exploring User Affective and Behavioral Responses to Social Media”. Please find below my comments and recommendations for the author:

1. The introduction of the papers should present more data to outline the importance of the research problem and research gap.

2. In the literature review section, the authors use many times general statements (e.g. Researchers have adopted...). I would recommend a more specific argumentation.

3. The „Theoretical background” section could be restructured to avoid repetition. The authors could present the variables' definitions and the evidence from the literature that supports the hypothesis. This could improve the flow of the paper.

4. In Figure 1, the font of the text should be adjusted for all the variables.

5. The authors should provide more information about the data collection procedure (e.g., the sampling technique).

6. I strongly recommend rewriting the following paragraph since it is unclear: „The measurements 374 for information, communication, and social overloads were specifically adapted from 375 [10]. The helpfulness value of social media is adapted and modified from the study [22]. 376 The measurement items of playfulness and social value are used from the study [23]. SM 377fatigue is measured based on a scale by [19]. SM engagement is assessed from the scale 378 [58]. Finally, SM loyalty is measured by four items based on the study [46], and SM dis- 379 continuous intention is assessed by three items based on [10]. The responses about dif- 380 ferent aspects of social media were measured on a 5-point Likert scale of 1 (strongly 381 disagree) to 5 (strongly agree).”

Indicate the authors that developed the scales. Also,  in Table 2. Measurement scales, reliability and validity assessment   - the authors should indicate the source of the scales.

6. The authors present Results mediation analysis, but the hypothesis did not anticipate this type of result. The authors argued the possible existence of direct effects, not indirect effects.

Comments on the Quality of English Language

Minor editing of English language required.

Author Response

Thank you very much for your thoughtful, detailed, and constructive suggestions, which have helped us significantly improve our manuscript. Please find the attached review notes and the manuscript. 

Thank you

Reviewer 2 Report

Comments and Suggestions for Authors

Dear Authors

I hope the following comments can help you improve your work's quality.

1.      I suggest you explain in the Introduction Section how your study will contribute to the existing body of knowledge and its practical implications.

2.      It is better to explain clearly in the Introduction Section what your study seeks to achieve and its expected contributions to the field. And clarify how and in which ways your findings help SM providers and practitioners. In Section 6.2, you presented some suggestions for SM providers and users. Do you think all your suggestions are feasible? Your suggestions do not address any direction and detailed information for solving the problems you found.

3.      In some places you used abbreviations (e.g., line 43: "WOM", line 177: "SNS", line 384: "KMO", and line 388: "EFA"). They need a full name.

4.      You proposed 8 hypotheses. Formulating at least one general research question or 8 specific research questions can better guide your research process.

5.      Lines 352-354 need reference.

66.      The presented information in the "sample and data collection Section" is inadequate. If others want to repeat your research, they need to know, for example, how and by whom your questionnaire was created, how you validated your questionnaire, how many and what kind of questions are in the questionnaire, who are the participants in the survey, how you found them, how you clarified the objectives of your work for them.  

7.      Does Table 2 present the questionnaire you used? Does it show all the questions you defined? If yes, why the number of questions is not even for each construct? Did you give the same rate to each question?  

8.      In Table 2, the questions of "helpfulness value" and "playfulness value" are the same.

9.      In lines 523-534 you claimed, "Unlike the previous studies which focused more on social media negative aspects and their effects on psychological strains, the present study discussed the SM resources and found that SM resources decrease SM fatigue". It should be noted that numerous studies emphasize the positive aspects of social media, with findings indicating that the advantages outweigh the disadvantages. For example, these studies highlight the growing number of users and the emergence of new social media platforms. How do you explain this contradiction?

Comments on the Quality of English Language

Minor editing of the English language is required.

Author Response

(The authors gave the same response as above.)

Round 2

Reviewer 2 Report

Comments and Suggestions for Authors

Dear Authors,

Thank you very much for the time and effort you invested in improving the quality of your work.

Comments on the Quality of English Language

Minor editing of the English language is required.